# A Cross-Sectional Study Investigating Canadian and Australian Adolescents’ Perceived Experiences of COVID-19: Gender Differences and Mental Health Implications

**DOI:** 10.3390/ijerph19074407

**Published:** 2022-04-06

**Authors:** Riana Marie, Audrey-Ann Journault, Rebecca Cernik, Paul Welch, Sonia Lupien, Brett McDermott, Joseph V. Moxon, Zoltan Sarnyai

**Affiliations:** 1Laboratory of Psychiatric Neuroscience, Centre for Molecular Therapeutics, Townsville, QLD 4811, Australia; riana.phillips@my.jcu.edu.au; 2Australian Institute of Tropical Health and Medicine, James Cook University, Townsville, QLD 4811, Australia; 3College of Medicine and Dentistry, James Cook University, Townsville, QLD 4811, Australia; paul.welch@jcu.edu.au (P.W.); brett.mcdermott@jcu.edu.au (B.M.); joseph.moxon@jcu.edu.au (J.V.M.); 4Centre for Studies on Human Stress, Montréal, QC H1N 3V2, Canada; audrey-ann.journault@umontreal.ca (A.-A.J.); rebecca.segall.cernik@umontreal.ca (R.C.); sonia.lupien@umontreal.ca (S.L.); 5Psychology Department, University of Montreal, Montréal, QC H3T 1J4, Canada; 6Psychiatry Department, University of Montreal, Montréal, QC H3T 1J4, Canada; 7College of Public Health, Medical, and Veterinary Sciences, James Cook University, Townsville, QLD 4811, Australia

**Keywords:** COVID-19, stress, school, mental health, adolescents

## Abstract

The coronavirus (COVID-19) disease pandemic has been associated with adverse psychological outcomes. This cross-cultural study (*N* = 1326, 71% female) aimed to investigate Canadian and Australian adolescents’ subjective experiences of COVID-19, gender differences, and psychological implications. Mixed-methods analyses were used to examine differences in COVID-19 experiences and mental health outcomes between country and gender in a Canadian (*N* = 913, 78% female) and an Australian sample (*N* = 413, 57% female) of adolescents. Canadian adolescents reported increased COVID-19 discussions and more concerns related to their COVID-19 experiences compared to Australian adolescents. Girls consistently reported more concerns related to COVID-19 and poorer psychological outcomes compared to boys. School lockdown for the Canadian sample may have played a role in these country differences. Further, girls might be at significantly more risk for mental health concerns during COVID-19, which should be considered in adolescent mental health initiatives during the pandemic. Although school disruption and separation of peers due to the pandemic likely have a role in adolescent perceived stressors and mental health, the differences between Canadian and Australian adolescents were less clear and future investigations comparing more objective pre-COVID-19 data to current data are needed.

## 1. Introduction

The coronavirus disease (COVID-19) pandemic is a global stressor with adverse health, psychological, and economic burdens [1,2]. To date, the World Health Organization (WHO) reports that there have been approximately 184 million reported cases and almost four million deaths associated with COVID-19 [3]. This disease has significantly affected individuals and communities who have experienced the loss of loved ones, health fears, future uncertainty, quarantine, social isolation, food/item insecurity, and business/school closures [2]. This transitional state of living has been related to poor mental health and well-being outcomes [4,5]. 

### 1.1. COVID-19 and Mental Health

From a review of past investigations of infectious diseases (e.g., SARS, Ebola, and H1N1 influenza), quarantine was associated with post-traumatic stress symptoms, confusion, and anger [6]. The full longitudinal extent of the consequences of the pandemic continues to be determined. One study found higher rates of anxiety, depression, substance use, and lower mental well-being among people in China compared to pre-COVID-19 norms [7]. The psychological impact of COVID-19 and quarantine experiences are still being investigated by living (frequently updated) systematic reviews [2,8]. Several studies have investigated the mental health and experiences of adolescents during the COVID-19 pandemic and past pandemics.

#### Adolescent Mental Health

Adolescence is a vulnerable developmental period associated with stressful transitions [9] and the onset of psychiatric disorders [10]. Therefore, the stress of a pandemic may increase the risk of mental health concerns within this age group [11]. COVID-19 has resulted in many school closures to reduce the transmission of the virus [12]. The absence of a structured school setting, disruption of routine, reduced social interactions, and general uncertainty may have psychological implications [13]. Increased levels of depression in the COVID-19-lockdown group were associated with factors such as smartphone and internet addiction [14]. With school closures, education strategies have transitioned to online environments. One study found that although some students were generally satisfied with online education, there was still a large proportion of students not comfortable with this form of learning [15]. Adolescent studies from multiple countries have found an increased risk of self-reported psychological symptoms, such as depression, anxiety, and stress associated with COVID-19 and lockdown situations [14,15,16,17,18,19,20,21,22,23,24]. Poorer psychological outcomes were more prevalent in girls [14,17,19,23,24]. Overall, the current study was conducted to contribute further knowledge about the effects of this pandemic using self-reported experiences of COVID-19 and mental health implications in a cross-cultural cohort of adolescents. 

### 1.2. Situating This Study in the Context of a Pandemic

It is important to situate the current research, as this cross-cultural study collected data from April to July 2020, a few months after the WHO declared COVID-19 to be a pandemic on March 11th, 2020 [25]. Data were collected in the province of Quebec, Canada and North Queensland, Australia, as these locations were where ethical clearance was attained. 

As transmissions increased, governments responded by enforcing restrictions, such as quarantines, social distancing, and mandatory mask-wearing, to reduce transmissions and mortality rates [3]. These restrictions varied according to country. At the time of testing in the province of Quebec (Canada, April to July), there were approximately 5500 (April) to 59,000 (July) confirmed cases, with approximately 100 (April) to 5700 (July) deaths (population ≈ 8.5 million) [26]. Conversely, in the state of Queensland (Australia), there had been a total of approximately 1100 confirmed cases and six deaths (population ≈ 5 million) [27]. Sanitary measures such as social distancing and hand sanitization were endorsed, with a 14-day obligatory quarantine if returning from outside of the country [28,29]. The main difference between the two countries was that in Canada, schools were closed, indoor/outdoor gatherings were forbidden, and non-essential businesses were closed [28]. In contrast, Australian schools and businesses were open [29]. 

Overall, Canada appeared to be experiencing greater economic and education burdens because of COVID-19. School closure may have a significant effect on the mental health of Canadian adolescents. As such, this cross-cultural investigation aimed to explore the differences between Canada and Australia, as well as gender, on adolescent experiences and concerns related to COVID-19. It should be noted that gender identity, rather than biological sex, was explored, as biological sex was not confirmed via the self-report scales; instead, students were asked for their gender identity (boy, girl, other). A secondary objective was to investigate the differences between country and gender on mental health outcomes within the context of the pandemic. It was hypothesized that Canadian adolescents would report greater concerns related to COVID-19 and poorer mental health outcomes. Consequently, it was also hypothesized that girls would report being more affected by COVID-19 and experiencing worse mental health outcomes than boys. Understanding cross-cultural differences in adolescent worries and mental health during COVID-19 may inform stress management education and policies during this global stressor. 

## 2. Materials and Methods

### 2.1. Participants

Adolescents (*N* = 1326, 71% female) were aged between 13 and 18 years old (*M* = 15.36 years, *SD* = 1.23 years). To be eligible for inclusion in the current study, participants had to be enrolled in high school (approximately 12 to 18 years of age). Canadian students were recruited online from private and public high schools in the province of Quebec from April to July 2020. Australian participants were recruited, using convenience sampling, from a private high school in North Queensland in July 2020. All participants included in this study gave their informed consent. Parental consent was not required for Canadian participants above 14 years old. Parental consent was obtained for Australian students under 16 years of age. This study was conducted in accordance with the Declaration of Helsinki. The Australian study was approved by the James Cook University Human Research Ethics Committee on 24 July 2019 (Ethics number: H7727). The Canadian study protocol was approved by the research ethics committee of the Centre intégré universitaire de santé et de services sociaux de l’Est de l’Île de Montréal on 17 April 2020. (Ethics number: 2019–1849.)

### 2.2. Demographics and COVID-19 Questionnaire

Demographics including age, gender, socioeconomic status, ethnicity, and household size were collected. Participants were asked questions related to COVID-19 experiences, worries, and concerns after demographic information was collected. These questions formed a questionnaire but were not derived from validated scales, as, at the time of testing, there were no validated COVID-19 measures due to being in the early stages of the pandemic. 

### 2.3. Psychological Scales

#### 2.3.1. Stress

The Perceived Stress Scale for Children (PSS-C [30]) is a 14-item measure of subjective stress, with scores ranging from 0 “never” to 3 “often”. The sum of the items (0–39, item 1 is not included due to being a practice question) was calculated for the current study. Higher scores indicate greater perceived stress levels in the past week (this deviates from the adult scale which examines perceived stress in the past month). The PSS-C was translated into French through double translation (two independent translations from English and French native speakers; see Appendix A for a more detailed description of this process) by the research team of the Centre for Studies on Human Stress (CSHS). The internal consistency (Cronbach’s alpha [α]) of this scale in the current study was α = 0.73 (Canada: α = 0.71 and Australia: α = 0.80). The Stress Mindset Measure—General (SMM-G [31]) is a recently developed 8-item measure of stress mindsets; the lens through which individuals view the effects of stress (positive versus negative). Scores can range from 0 “strongly disagree” to 4 “strongly agree”, with higher scores indicating more positive stress mindsets (scores 2+ fall on the more “stress-is-enhancing” spectrum and scores <2 fall on the more “stress-is-debilitating” spectrum). Mean scores for this scale were reported. A French version of the scale was developed for the Canadian sample, through double translation by the CSHS research team. The internal consistency of this scale was α = 0.78 (Canada: α = 0.82 and Australia: α = 0.67).

#### 2.3.2. Anxiety 

The Childhood Anxiety Sensitivity Index (CASI [32]) is an 18-item scale measuring an individual’s sensitivity to anxiety or innate beliefs that anxiety can have harmful consequences. Scores range from 1 “not at all” to 3 “a lot”. The sum of all the items was calculated (18–54), with higher scores suggesting an increased sensitivity to anxiety. A validated French version was used for the Canadian sample [33]. The internal consistency of this scale was α = 0.87 (Canada: α = 0.86 and Australia: α = 0.91). The State-Trait Anxiety Inventory for Children—State Subscale (STAIC-S [34]) is a 20-item measure of momentary anxiety, with scores ranging from 1 “very [emotion, e.g., “worried”]” to 3 “not [emotion, e.g., worried]”. The sum of all items (20–60) was calculated, with higher scores suggesting increased levels of momentary anxiety. The validated French version of this inventory was used in the Canadian sample [35]. The internal consistency of this scale was α = 0.90, (Canada: α = 0.90 and Australia: α = 0.92). The Children’s Test Anxiety Scale (CTAS [36]) is a 25-item scale that measures the level of performance anxiety a participant might experience during a test. Scores can range from 1 “almost never” to 4 “almost always”. The sum of all items (25–100) was calculated, with higher scores indicating increased levels of test anxiety. The CTAS was translated into French through double translation by the CSHS research team. The internal consistency of this scale was α = 0.94 (Canada: α = 0.94 and Australia: α = 0.95).

#### 2.3.3. Co-Rumination

The Co-Rumination Questionnaire (CRQ) is a short-form (9-item) scale extracted by Arroyo [37] of the original 27-item Co-Rumination Questionnaire [38], which measures how often a participant dwells on negative situations with their friends; for example, “We spend most of our time together talking about problems that my friend or I have”. Scores can range from 1 “not true at all” to 5 “very true”. The sum of all items (9–45) was calculated, with higher scores suggesting that participants engaged in a greater amount of co-rumination with their friends. The CRQ was translated into French through double translation by the CSHS research team. The internal consistency of this scale was α = 0.88 (Canada: α = 0.87 and Australia: α = 0.91).

#### 2.3.4. Depression

The Beck Depression Inventory-II (BDI-II [39]) is a 21-item scale that was used, after modification based on the French BDI [40], to assess the intensity and severity of depressive symptomatology in the Canadian sample. This scale has four responses corresponding to a score between 0 and 3 indicating the severity of the symptom. Higher scores indicate greater depression severity. The internal consistency of this scale was α = 0.91 for the Canadian sample. The Patient Health Questionnaire-9 for Adolescents (PHQ-9A [41], adapted for youth by Johnson et al. [42]) is a 9-item scale that was used to measure depression severity within the Australian sample. Scores can range from 0 “not at all” to 3 “nearly every day”, with higher scores suggesting that the participant may be experiencing a greater number of depression symptoms. The internal consistency of this scale was α = 0.90 for the Australian sample.

### 2.4. Procedure

The data used in this between-subjects cross-sectional investigation were collected at baseline (Canada: April to June 2020 and Australia: July 2020) before the implementation of a stress mindset intervention designed to promote “stress-is-enhancing” mindsets (to be part of another paper). These baseline data were collected using an online survey via the Qualtrics platform (Qualtrics, Provo, UT, USA; July 2020 version [43]). The online survey consisted of demographic questions, the COVID-19 questionnaire, and randomised validated self-report scales. Participants spent approximately 30 min completing this survey. The Canadian study was completed in French, while the Australian study was completed in English. It is important to disclose that the protocol used in the current study was adapted from Journault and Lupien [44] and a detailed description of the original protocol can be found at https://osf.io/u4cmf (accessed on 27 March 2022).

### 2.5. Statistical Analyses

Analyses were conducted using Statistical Package for the Social Sciences (SPSS v25). Normality was assumed based on the sufficient sample size (*N* = 1326) and acceptable skewness and kurtosis ranges (scores within +2/−2 and +7/−7 [45,46]), and a visual inspection of quantile-quantile (Q–Q) plots. Missing cases included outliers (only cases outside of scale limits were removed as outliers [*n* = 5]) and incomplete survey responses. The independent variables in this investigation were country (Canada and Australia) and gender (boys and girls). Descriptive statistics are presented as the means and 95% confidence intervals (*CI*). For the COVID-19 questionnaire, continuous data were analysed using a 2 (country) × 2 (gender) analysis of variance (ANOVA) model. Multivariate ANOVAs (MANOVAs) were used to assess questions that had more than one continuous outcome. Effect sizes for the MANOVAs were reported as partial eta squared (*η_p_*^2^) and based on interpretations provided by Cohen [47]: 0.02 (small effect), 0.06 (moderate effect), and 0.14 (large effect). Chi-squared (χ^2^) tests were performed for categorical variables. Corresponding effect sizes were reported as Goodman and Kruskal’s tau (τ) and Cramer’s V based on interpretations reported by Kim [48]. Bonferroni-adjusted simple effects for the categorical variables were performed using Z-tests. For the psychological scales, 2 (country) × 2 (gender) ANOVAs were used to investigate the main effects and interaction effects. Bonferroni-adjusted simple effects analyses were performed for significant interaction effects with Cohen’s d reported for effect size [47]. The scales used to measure depression severity were transformed into z-scores to compare differences between countries using the two different depression scales (BDI-II and PHQ9-A). Differences were considered significant for all analyses at *p* < 0.05. An exploratory inter-item analysis was conducted on the PSS-C scale items, using independent samples *t*-tests, with *p*-values of < 0.01 indicating significance due to multiple comparisons. 

## 3. Results

### 3.1. Sample Characteristics 

The Canadian sample consisted of 913 French-speaking students (78% female) from both private and public high schools in Quebec, Canada (*M*_age_ = 15.41 years, *SD*_age_ = 1.08). The Australian sample consisted of 413 English-speaking students (57% female) from a private high school in Queensland, Australia (*M*_age_ = 15.24 years, *SD*_age_ = 1.50). The Canadian sample identified mostly as white/Caucasian (65%). The remainder of the sample identified as 6% multiracial/other, 3% Asian origin, 3% Middle Eastern or North African, 1.8% Caribbean origin, 1.4% Black origin, 1% Central and Southern America, and 0.3% First Nations. Similarly, the Australian sample identified mostly as white/Caucasian (70%). The remainder of the sample identified as 15% Asian, 12% multiracial/other, 1% African, and 1% Australian Aboriginal or Torres Strait Islander. Both samples reported being from predominantly “quite comfortable” financial backgrounds (Canada: 51%, Australia: 48%) and living with approximately 5 people in their household at the time of testing.

### 3.2. COVID-19 Questionnaire

See Appendix A for the descriptive statistics related to the COVID-19 questionnaire.

#### 3.2.1. Frequency and Impact of Symptoms Resembling COVID-19

Australian adolescents reported experiencing significantly more physical symptoms resembling COVID-19 compared to Canadian adolescents (*F*_(1, 1285)_ = 13.226, *p* < 0.001, *η_p_^2^* = 0.010, small effect). Girls reported experiencing COVID-19-like symptoms significantly more than boys (*F*_(1, 1285)_ = 4.143, *p* = 0.042, *η_p_^2^* = 0.003, small effect). There was no significant interaction effect between countries and gender (*p* > 0.05) on frequency of symptoms. Girls reported that the COVID-19-like symptoms they experienced had affected their life significantly more compared to boys (*F*_(1, 784)_ = 4.565, *p* = 0.033, *η_p_^2^* = 0.006, small effect). However, there was no significant interaction effect between countries and gender (*p* > 0.05) on the impact of symptoms.

#### 3.2.2. COVID-19 Discussions

Canadian adolescents were significantly more likely to discuss COVID-19 compared to Australian adolescents (*χ*^2^ _(4, *N* = 1300)_ = 69.839, *p* < 0.001, two-tailed, *V* = 0.232, medium effect, τ = 0.054). In general, girls were significantly more likely to discuss COVID-19 more than boys (*χ*^2^ _(4, *N* = 1300)_ = 36.263, *p* < 0.001, two-tailed, *V* = 0.167, medium effect, τ = 0.028). Bonferroni-adjusted *Z*-tests found that Canadian girls were significantly more likely to discuss COVID-19 approximately 2 to 5 times per day (*p* < 0.05). In contrast, Canadian boys were significantly more likely to only discuss COVID-19 a few times a week (*p* < 0.05). 

#### 3.2.3. Media Use

Canadian adolescents were significantly more likely to report following the news about COVID-19 than Australian adolescents (*χ*^2^ _(1, *N* = 1291)_ = 22.958, *p* < 0.001, two-tailed, *V* = 0.133, small effect, τ = 0.018). Girls in general were significantly more likely to report following the news about COVID-19 compared to boys (*χ*^2^ _(1, *N* = 1291)_ = 25.867, *p* < 0.001, two-tailed, *V* = 0.142, small effect, τ = 0.020). Bonferroni-adjusted *Z*-tests found that Canadian girls were significantly more likely to follow the news about COVID-19 (*p* < 0.05). Canadian adolescents reported consulting traditional media for news (e.g., newspaper, television, or radio) significantly more than Australian adolescents (*F*_(1, 1230)_ = 17.355, *p* < 0.001, *η_p_^2^* = 0.014, small effect). Girls reported consulting traditional (*F*_(1, 1230)_ = 6.007, *p* = 0.014, *η_p_^2^* = 0.005, small effect) and social media (*F*_(1, 1230)_ = 8.739, *p* = 0.003, *η_p_^2^* = 0.007, small effect) for news significantly more than boys. There was no significant difference between countries and genders in how often participants consulted online websites for news (*p* > 0.05).

#### 3.2.4. Stress and Concerns Related to COVID-19

Canadian adolescents reported experiencing significantly more stress before COVID-19 (*F*_(1, 1278)_ = 17.465, *p* < 0.001, *η_p_^2^* = 0.013, small effect), at the time of testing (*F*_(1, 1278)_ = 10.427, *p* = 0.001, *η_p_^2^* = 0.008, small effect), and related to COVID-19 (*F*_(1, 1278)_ = 201.893, *p* < 0.001, *η_p_^2^* = 0.136, medium effect) than Australian adolescents. Further, girls in both samples reported experiencing significantly more stress before COVID-19 (*F*_(1, 1278)_ = 132.496, *p* < 0.001, *η_p_^2^* = 0.094, medium effect), at the time of testing (*F*_(1, 1278)_ = 110.931, *p* < 0.001, *η_p_^2^* = 0.080, small effect), and related to COVID-19 (*F*_(1, 1278)_ = 14.456, *p* < 0.001, *η_p_^2^* = 0.011, small effect) compared to boys. There were no significant interaction effects for any of these variables (*p* > 0.05). A repeated-samples ANOVA (country × time) found no significant difference between reported stress levels prior to COVID-19 and at the time of testing for both countries (*F*_(1, 1280)_ = 3.710, *p* = 0.054, *η_p_^2^* = 0.003, small effect). 

Canadian adolescents reported experiencing significantly more concern about their personal health (*F*_(1, 1280)_ = 9.959, *p* = 0.002, *η_p_^2^* = 0.008, small effect), the health of their parents (*F*_(1, 1280)_ = 40.487, *p* < 0.001, *η_p_^2^* = 0.031, small effect), and the health of a loved one (*F*_(1, 1280)_ = 36.501, *p* < 0.001, *η_p_^2^* = 0.028, small effect) compared to Australian adolescents. Canadian adolescents were significantly more concerned about the continuation of the school year (*F*_(1, 1280)_ = 138.906, *p* < 0.001, *η_p_^2^* = 0.098, medium effect) compared to Australian adolescents. However, there was no significant country differences in concerns about personal/parental job security (*p* > 0.05) or access to items (*p* > 0.05). Girls reported experiencing significantly more concern about their personal health (*F*_(1, 1280)_ = 27.188, *p* < 0.001, *η_p_^2^* = 0.021, small effect), the health of their parents (*F*_(1, 1280)_ = 8.158, *p* = 0.004, *η_p_^2^* = 0.006, small effect), and the health of a loved one (*F*_(1, 1280)_ = 18.800, *p* < 0.001, *η_p_^2^* = 0.014, small effect) compared to boys. Girls were also significantly more concerned about the continuation of the school year (*F*_(1, 1280)_ = 58.109, *p* < 0.001, *η_p_^2^* = 0.043, small effect), personal/parental job security (*F*_(1, 1280)_ = 16.225, *p* < 0.001, *η_p_^2^* = 0.013, small effect), and accessibility of items (*F*_(1, 1280)_ = 15.618, *p* < 0.001, *η_p_^2^* = 0.012, small effect) than boys. However, there were no significant interaction effects between country and gender for any of these variables (*p* > 0.05). 

### 3.3. Adolescent Mental Health in the Context of COVID-19

See Table 1 for the descriptive statistics related to the mental health outcomes and Appendix A for the inferential statistics. 

#### 3.3.1. Stress

Australian adolescents reported experiencing significantly more perceived stress compared to Canadian adolescents (*F*_(1, 1179)_ = 8.438, *p* = 0.004, *η_p_^2^* = 0.007, small effect). Further, girls in both samples reported experiencing significantly more perceived stress compared to boys (*F*_(1, 1179)_ = 40.105, *p* < 0.001, *η_p_^2^* = 0.033, small effect). There was no significant interaction between country and gender on the PSS-C (*p* > 0.05). Canadian adolescents reported significantly more “stress-is-debilitating” mindsets compared to the Australian adolescents (*F*_(1, 1225)_ = 19.789, *p* < 0.001, *η_p_^2^* = 0.016, small effect). There were no significant effects for gender (*p* > 0.05) or the interaction between country and genders (*p* > 0.05) on the SMM-G.

#### 3.3.2. Anxiety

Australian adolescents reported experiencing significantly more test anxiety compared to Canadian adolescents (*F*_(1, 1133)_ = 9.057, *p* = 0.003, *η_p_^2^* = 0.008, small effect). Girls reported being significantly more sensitive to anxiety (*F*_(1, 1222)_ = 135.327, *p* < 0.001, *η_p_^2^* = 0.100, medium effect), as well as experiencing significantly more state anxiety (*F*_(1, 1207)_ = 57.198, *p* < 0.001, *η_p_^2^* = 0.045, small effect) and test anxiety (*F*_(1, 1133)_ = 79.811, *p* < 0.001, *η_p_^2^* = 0.066, medium effect) compared to boys. There were no other significant effects for country in state anxiety or the interaction between country and genders for both test and state anxiety (*p* > 0.05).

#### 3.3.3. Co-rumination

Canadian adolescents reported co-ruminating with others significantly more than Australian adolescents (*F*_(1, 1152)_ = 6.616, *p* = 0.010, *η_p_^2^* = 0.006, small effect). Girls reported co-ruminating significantly more than boys (*F*_(1, 1152)_ = 76.420, *p* < 0.001, *η_p_^2^* = 0.062, medium effect). There was a significant interaction between country and gender (*F*_(1, 1152)_ = 9.208, *p* = 0.002, *η_p_^2^* = 0.008, small effect). Canadian boys reported engaging in co-rumination significantly more compared to Australian boys (*p* = 0.001, *d* = 0.36, small effect).

#### 3.3.4. Depression

Girls showed significantly more signs of depression compared to boys (*F*_(1, 1161)_ = 68.834, *p* < 0.001, *η_p_^2^* = 0.056, small effect). There were no other significant effects between country and gender (*p* > 0.05) on the *z*-scored depression scales. The percentage of students scoring above the clinical cut-off scores is illustrated in pie charts in Figure 1.

### 3.4. Exploratory Inter-Item Analysis on Perceived Stress

Based on the discrepancy between countries on reported levels of stress within the COVID-19 implementation variables and the PSS-C, additional exploratory analyses were conducted on the PSS-C to further examine the scale items (see Appendix A for scale items and descriptive statistics). Australian adolescents felt significantly more rushed/hurried (*t*_(1185)_ = −4.136, *p* < 0.001, *d* = 0.27, small effect), that they did not have enough time to do what they wanted (*t*_(1185)_ = −10.354, *p* < 0.001, *d* = 0.66, medium effect), were worried about being too busy (*t*_(1185)_ = −4.069, *p* < 0.001, *d* = 0.26, small effect), and did not feel that they were getting enough sleep (*t*_(1183)_ = −4.793, *p* < 0.001, *d* = 0.31, small effect) compared to Canadian adolescents. In contrast, Canadian adolescents reported being significantly more angry (*t*_(1183)_ = 3.374, *p* < 0.001, *d* = 0.22, small effect) and not being able to spend time with their friends (*t*_(1183)_ = 15.372, *p* < 0.001, *d* = 0.99, large effect) compared to Australian adolescents. See Figure 2 for the differences between scale items.

## 4. Discussion

This cross-cultural investigation aimed to explore adolescent experiences and concerns related to COVID-19 between countries and gender. A secondary objective was to investigate the differences between country and gender on mental health outcomes within the context of the pandemic.

The hypothesis that adolescents from Canada would report being more affected by the pandemic circumstances and have worse psychological outcomes compared to the Australian adolescents was partially supported. Canadian adolescents did engage in more discussions about COVID-19 and followed the news, particularly traditional media, more than Australian adolescents. Canadian adolescents also reported experiencing more stress at the time of testing, before COVID-19, and as a consequence of COVID-19 compared to Australian adolescents. Finally, Canadian adolescents reported more concerns about health (personal, parental, and loved ones) and the continuation of the school year, suggesting that the Canadian sample were experiencing more worries and concerns associated with COVID-19. This may have been the result of the stricter lockdown conditions in Canada. Lockdown conditions have been associated with increased concerns and negative mental health trajectories [11,13,16,20,51]. Unlike Australia, which is an island nation, Canadian adolescents may have also been reporting greater COVID-19 effects, as a result of sharing land boundaries with the United States of America, a country experiencing rising rates of transmission [52]. Further, increased media consumption of COVID-19-related information was associated with increased worry, but, in turn, more preventative behaviors [53]. Therefore, increased concerns and worries in the Canadian sample could be associated with their increased media consumption.

Conversely, the Australian sample reported experiencing significantly more physical symptoms resembling COVID-19 compared to Canada, at the time of testing. This finding could be associated with seasonal differences at the time of testing or the increased media consumption prompting more preventative hygiene practices in the Canadian sample [53]. Interestingly, adolescents from both countries perceived no difference in their stress levels before COVID-19 and at the time of testing. Although this could be a sign of stress resilience (e.g., acceptance or growth under stress) or difficulty introspecting stress levels from the past, it would be worth comparing more objective and physiological measures of stress from before COVID-19 to now, rather than just relying on these subjective reports.

Although Canadian adolescents reported experiencing more stress at the time of testing on the COVID-19 questionnaire, Australian adolescents reported more perceived stress on the PSS-C. It is unclear why we found contradictory findings and although it may be due to the psychometric difference between validated psychological measures compared to a simple questionnaire, it could also be associated with the novel pandemic circumstances or the month of testing. Australian adolescents were tested in July, four months after the pandemic was declared, compared to Canadian adolescents who were tested from a month after the announcement. The PSS-C contains several questions that may not be particularly relevant to adolescents experiencing strict COVID-19 restrictions (e.g., time spent playing with friends). Australian adolescents felt significantly more rushed/hurried, that they did not have enough time to do what they wanted, were worried about being too busy, and did not feel that they were getting enough sleep compared to Canadian adolescents. Canadian adolescents reported being significantly angrier and there was also a large difference in the lack of time spent with friends than Australian adolescents. These responses could be associated with the fact that Australian adolescents were at school full time, whereas Canadian adolescents were in lockdown, not at school, and with very limited face-to-face contact with friends. Similarly, Australian adolescents reported experiencing more test anxiety, which again could be associated with the school environment. From 2020 to 2021, the MyStrengths Youth Mental Health Survey found that schoolwork was the biggest stressor for Australian high school students [54], which may align with the increased perceived stress and test anxiety in Australian adolescents. However, the novel COVID-19-related school closures may explain the reports of increased worries, anger, and lack of time spent with friends within the Canadian sample report, as the absence of a structured school setting, disruption of routine, reduced social interactions, and general uncertainty may still have psychological implications [13]. Further, online learning may not be appropriate for some students, as it has been previously reported that some do not find this type of learning effective [15].

Finally, Canadian adolescents reported viewing stress as more debilitating and co-ruminating more with friends than Australian adolescents. Having a more “stress-is-debilitating” mindset may be associated with less proactive coping strategies (e.g., ruminating, withdrawal, and avoidance) [31]. The increased co-rumination in Canadian adolescents aligns with the more frequent discussions about COVID-19 reported earlier. Although school closure may have limited face-to-face contact with peers, online environments offer novel avenues for social connectedness [55]. Overuse of online media and communication can be linked to poor mental health outcomes, increased co-rumination, online bullying, social isolation, and the spread of misinformation [14,55]. Therefore, it may be worth considering the time adolescents spend in online environments, as a potential health-risk behavior in future investigations.

The hypothesis that the effect of COVID-19 would be greater in girls was supported. Girls reported more discussions about COVID-19 and following the news, particularly traditional and social media. They also reported experiencing more symptoms and felt that these symptoms had a bigger impact compared to boys. Girls reported more subjective stress before, during, and related to COVID-19 than boys. Further, girls were more concerned about health, school continuation, personal/parental job security, and item accessibility. Finally, the hypothesis that girls would be experiencing poorer mental health outcomes compared to boys was supported. Girls reported experiencing more perceived stress, anxiety sensitivity, state anxiety, test anxiety, co-rumination, and depression symptoms. Greater effect sizes were also consistently observed for girls in self-assessed COVID-19 experiences and the psychological outcomes, particularly anxiety sensitivity. This finding is consistent with the increased risk of depression, anxiety, and co-rumination symptoms seen in girls during adolescence [56,57].

Although the causes are likely multifaceted, the experience of more stressors and pubertal changes early during adolescence may be associated with this difference between girls and boys [56]. It could also be associated with socialization or gender roles during child-rearing stages, where girls are encouraged to disclose their feelings and consequently answer accordingly on psychological scales assessing mental health [58]. In contrast, boys may be discouraged from sharing their feelings, as it does not align with traditional masculine gender roles and often experience externalizing symptoms such as aggression [58]. Closeness with parents appeared to moderate the long-term effects of these challenges [53]. There is further evidence to suggest that this gender difference persists during COVID-19 [14,17,19,23,24]. Adolescent girls also appear to experience more self-assessed health concerns in combination with psychological morbidity [59], which could explain the increased frequency and impact of reported COVID-19-related symptoms in the current study. Therefore, girls report more concerns related to COVID-19 and internalizing mental health symptoms, which should be considered in COVID-19-related mental health policies or interventions. However, this should not exclude boys from consideration in policies, as the current study measures may not have captured externalizing symptoms of mental health challenges, such as aggression. Future research is warranted exploring the association between gender roles and psychological outcomes during COVID-19.

### Limitations

This study was limited by the use of self-report measures, which can sometimes result in biases, such as socially desirable answers or difficulty with introspection [60], which could explain our contradictory findings in perceived stress. Further, the COVID-19 questionnaire was not a validated scale. Although significant, the majority of the reported differences were only small in size, particularly when investigating country differences. Due to the cross-sectional design, we are unable to establish causation. Therefore, pre-COVID-19 data would be critical to fully determine whether COVID-19 has harmed mental health and well-being during adolescence. It is possible that due to the higher proportion of females in the Canadian sample, the differences between countries may also be confounded by female-related mental health risks. A limitation of the Australian sample was that the participants were recruited from a single private school rather than the multiple sites as in the Canadian sample. The diversity of both the Canadian and Australian samples was limited, with most students identifying as Caucasian and financially “quite comfortable”. Finally, differences in language may have also played a role in the differences in responses between countries.

## 5. Conclusions

In summary, compared to males, female adolescents from Australia and Canada reported more symptoms of stress, anxiety, and depression during the first months of the COVID-19 pandemic and reported more personal concerns and worries associated with the pandemic. Mental health responses during the pandemic should consider this significant gender difference and the effect of gender roles when developing stress management programs, as girls and boys may experience different responses to certain stressors. Disrupting school and peer interactions could be associated with the increased concerns about the continuation of the school year and being able to spend time with friends in the Canadian sample, compared to the Australian sample who remained at school. Further investigations examining changes before and after COVID-19 and scale appropriateness within the pandemic context are necessary to fully understand the consequences of COVID-19 during adolescence.

## Figures and Tables

**Figure 1 ijerph-19-04407-f001:**
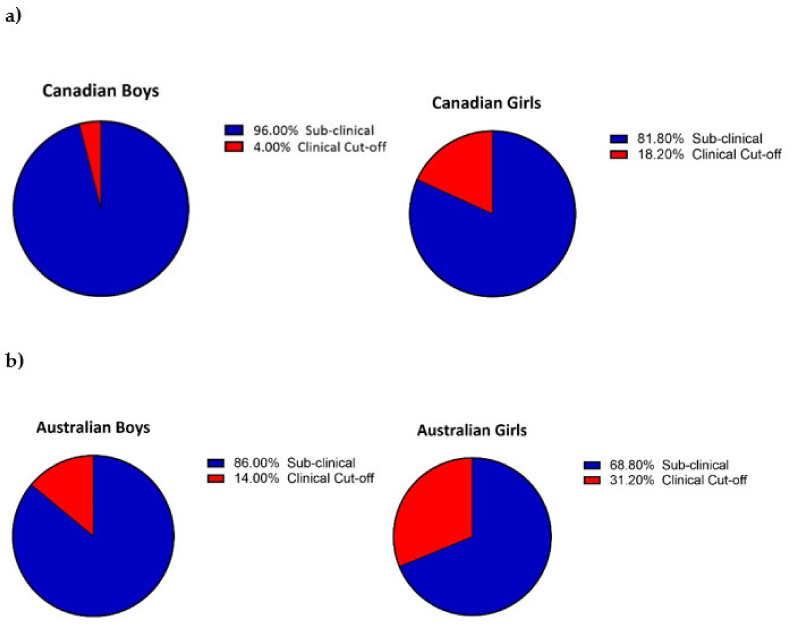
Depression Severity of Adolescents Stratified by Gender. *Note.* (**a**) Percentage (%) of Canadian boys and girls in the current study scoring equal to or above the clinical cut-off score of 23 [49] on the BDI-II. (**b**) Percentage (%) of Australian boys and girls in the current study scoring equal to or above the clinical cut-off score of 11 [50] on the PHQ9-A.

**Figure 2 ijerph-19-04407-f002:**
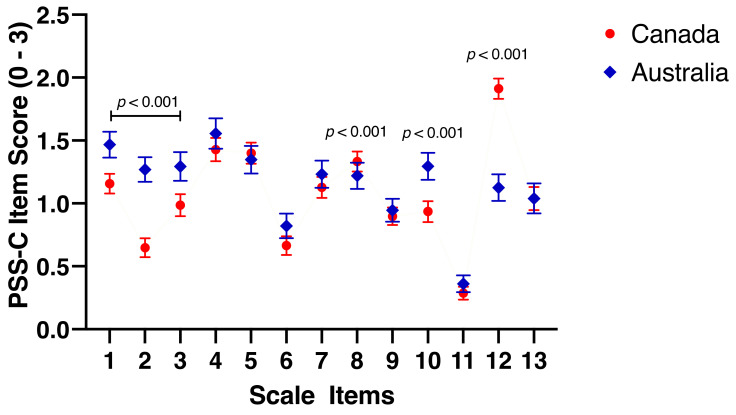
Inter-item Analysis of the PSS-C. *Note.* Mean (95% *CI*) differences between countries on the PSS-C scale items.

**Table 1 ijerph-19-04407-t001:** Mean (95% CI) for Psychological Scales Stratified by Country and Gender.

**Variable**	**Canada** **(*N* = 913)**		**Australia** **(*N* = 413)**		**Country *p*-Value**	**Gender *p*-Value**
	**Boys** **(*n* = 198)**	Girls (*n* = 708)	Boys (*n* = 172)	Girls (*n* = 234)		
**PSS-C**	**12.21 (11.32, 13.11)**	15.39 (14.93, 15. 85)	13.94 (12.93, 14.95)	16.13 (15.28, 16.98)	**0.004**	**<0.001**
*N (Missing)*	*178 (20)*	*667 (41)*	*140 (32)*	*198 (36)*		
**SMM-G**	1.61 (1.52, 1.70)	1.54 (1.50, 1.59)	1.81 (1.70, 1.92)	1.75 (1.66, 1.84)	**<0.001**	0.167
*N (Missing)*	*197 (1)*	*689 (19)*	*141 (31)*	*202 (32)*		
**CASI**	27.51 (26.54, 28.49)	32.79 (32.27, 33.30)	28.05 (26.92, 29.18)	33.75 (32.78, 34.71)	0.113	**<0.001**
*N (Missing)*	*192 (6)*	*691 (17)*	*144 (28)*	*198 (36)*		
**STAIC-S**	32.36 (31.38, 33.34)	36.22 (35.71, 36.73)	32.07 (30.94, 33.20)	35.30 (34.36, 36.25)	0.200	**<0.001**
*N (Missing)*	*188 (10)*	*682 (26)*	*140 (32)*	*201 (33)*		
**CTAS**	50.37 (47.90, 52.83)	58.93 (57.66, 60.21)	52.15 (49.45, 54.85)	64.04 (61.75, 66.33)	**0.003**	**<0.001**
*N (Missing)*	*169 (29)*	*631 (77)*	*141 (31)*	*196 (38)*		
**CRQ**	26.68 (25.55, 27.82)	29.69 (29.12, 30.27)	23.73 (22.45, 24.98)	29.94 (28.89, 30.98)	**0.010**	**<0.001**
*N (Missing)*	*170 (28)*	*647 (61)*	*139 (33)*	*200 (34)*		
**Depression Z Scores (BDI-II, PHQ9-A)**	−0.43 (−0.57, −0.28)	0.11 (0.04, 0.19)	−0.35 (0.51, −0.19)	0.24 (0.10, 0.37)	0.141	**<0.001**
*N (Missing)*	*173 (25)*	*652 (56)*	*137 (35)*	*203 (31)*		

*Note*. Continuous data are presented at *M* ± 95% *CI. N/n* = sample size. (*Missing*) = number of missing cases. Bold *p*-values indicate a statistically significant outcome.

## Data Availability

The Canadian study was formally pre-registered: https://osf.io/u4cmf, accessed on 7 February 2022. However, the Australian experiment reported in this article was not formally pre-registered. Neither the data nor the materials for the Australian study have been made available on a permanent third-party archive and cannot be shared, as although informed consent was attained for general de-identified summaries, the participants were not asked for consent to share their individual data points in open-access datasets.

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
