# Peer review of "A Cross-Sectional Study Investigating Canadian and Australian Adolescents’ Perceived Experiences of COVID-19: Gender Differences and Mental Health Implications"

_ijerph, 2022, doi:10.3390/ijerph19074407_

Round 1

Reviewer 1 Report

The manuscript is well designed and very easy to read.

In my opinion, the introduction section describing the characteristics of the Pandemic in Australia and Canada is too long. Perhaps it should be shortened or moved to an appendix of supplementary material.

Why has the study been conducted in Australia and Canada and not in New Zealand and the United States? Please include a brief justification.

The material and methods section states how many adolescents participated, but it is not clear the selection mechanism is nowhere to be found. It should be explained in detail.

There appears to be a contradiction between the material and methods section, which states that informed consent was obtained from all participants. The statement on lines 513 and 514 says that consent was not obtained from the participants.

One issue is that it discusses the gender of the students. Gender is not a politically correct synonym for sex. Was the sex (a biological question) asked, or were the participant's sexual orientation and roles asked ( male, female, non-binary, LGTBI, etc.)  

The material and methods section indicates that the sample in Canada was conducted in French. In one of the scales, it is stated that double-blind translation was used. This is not a well-known procedure in public health where back translation, or the WHO translation system, is more commonly used. The process should be explained to readers who are not familiar with it.

For the other scales, no translation procedure is indicated. Whether the questionnaires are validated in French and the reference provided should be revealed.

Crammer's V test is used to measure the effect size in contingency tables. The problem with Crammer's V statistic is that it is based on the Chi-square, so if it is very large, it would also be large. I suggest using SPSS to estimate the Lambda statistic. The latter is calculated with the same crosstabs procedure as Crammer's V.

I assume that the alpha value refers to Cronbach alpha. Please put it to facilitate the reading for a reader not very familiar with the methodological aspects.

It might be interesting to add the questionnaires used in English as supplementary material.

Author Response

The authors thank the reviewers for their constructive feedback on the manuscript.

Reviewer 1

The manuscript is well designed and very easy to read.

  1. In my opinion, the introduction section describing the characteristics of the Pandemic in Australia and Canada is too long. Perhaps it should be shortened or moved to an appendix of supplementary material.

The authors have restructured the introduction to improve the conciseness of the paper. More detailed information about the pandemic was moved to the supplementary material.

Section: Introduction

Line numbers: 80 – 81, 83 – 84, 86, 682 - 710

  1. Why has the study been conducted in Australia and Canada and not in New Zealand and the United States? Please include a brief justification.

The authors’ ethical clearance was specific to the locations of their affiliations and, as such, countries outside of their affiliation were not investigated. It would be interesting to pursue cross-cultural variations of this study in the future. We have included a brief rationale in the introduction to clarify this point.

Section: Introduction

Line numbers: 74 - 74

  1. The material and methods section states how many adolescents participated, but it is not clear the selection mechanism is nowhere to be found. It should be explained in detail.

The selection criteria for participant recruitment has been included by the authors in the participant section within the Materials & Methods.

Section: Materials & Methods

Line numbers: 105 - 109

  1. There appears to be a contradiction between the material and methods section, which states that informed consent was obtained from all participants. The statement on lines 513 and 514 says that consent was not obtained from the

Thank you for bringing this to the authors’ attention, this was an error in writing style. The authors did have consent from all participants to participate in the Canadian and Australian arms of the study and have their results reported generally in a de-identified manner. However, the Australian informed consent forms did not cover open databases and, as such, the Australian participants did not consent to share their individual data points within open databases. This has been amended in the text.

Section: Informed consent

Line numbers: 509 - 511

  1. One issue is that it discusses the gender of the students. Gender is not a politically correct synonym for sex. Was the sex (a biological question) asked, or were the participant's sexual orientation and roles asked (male, female, non-binary, LGTBI, etc.)  

The authors understand the importance of ensuring that politically correct terminology is used within manuscripts. “Gender” was selected to refer to gender identity rather than sex or sexual orientation, as it is the more appropriate term according to the APA guidelines (see below) and the method in which the data was collected. We did not verify biological sex or their sexual orientation, but asked whether they were a boy, girl or other, which aligns with gender identity. This was discussed in the introduction and further clarified in these revisions.

Section: Introduction

Line numbers: 91 – 93

  1. The material and methods section indicates that the sample in Canada was conducted in French. In one of the scales, it is stated that double-blind translation was used. This is not a well-known procedure in public health where back translation, or the WHO translation system, is more commonly used. The process should be explained to readers who are not familiar with it.

The authors have included more detail in the manuscript to explain the double-translation process. In the text, the term “double-blind” was used and this should instead be “double-translation”, which has been amended in-text. The double-translation process requires two individuals: both need to be bilingual in both English and French, but one person’s native language (ie. mother tongue) is French and the other person’s native language is English. The original English version of the questionnaire is translated from English to French by the francophone person. In other words, they only see the English items and they must translate them to French. Then, the anglophone receives the document but only sees the French translated items (and thus, do not see the original English items). This person’s job is to translate the French items back to English. At the end of this process, both individuals compare the original English items and the items that were produced using the double-translation technique. Doing so ensures that the French items were successfully translated. If the meaning of an item is not exact/could be improved, both individuals come to a consensus on how the item could be worded otherwise in order to obtain a better translation of the original item. This method has been used to translate the PISA survey: https://www.oecd.org/pisa/sitedocument/PISA-2015-Technical-Report-Chapter-5-Translation.pdf

Section: Materials & Methods

Line numbers: 131 – 133, 704 – 717  

  1. For the other scales, no translation procedure is indicated. Whether the questionnaires are validated in French and the reference provided should be revealed.

We have now clarified whether each scale was double-translated or a French validated version in-text.

Section: Materials & Methods

Line numbers: 142, 148 – 149, 154 – 155, 160, 170, 174 – 175, 603 – 608

  1. Crammer's V test is used to measure the effect size in contingency tables. The problem with Crammer's V statistic is that it is based on the Chi-square, so if it is very large, it would also be large. I suggest using SPSS to estimate the Lambda statistic. The latter is calculated with the same crosstabs procedure as Crammer's V.

Using the same procedure, we were met with the following error: “cannot be computed because the asymptotic standard error equals zero” for the Lambda statistic. The authors are not familiar with the statistic and if the reviewer had any further recommendations regarding this statistic and its inclusion in the paper, it would be highly appreciated. The only directional measure that was computed was the Goodman and Kruskal’s tau, which we believe might be similar to Goodman and Kruskal’s lambda and have therefore included it, in addition to V, in the paper.          

Section: Materials & Methods, Results

Line numbers: 210, 251, 253, 260, 262                                      

  1. I assume that the alpha value refers to Cronbach alpha. Please put it to facilitate the reading for a reader not very familiar with the methodological aspects.

The alpha value does refer to Cronbach’s alpha and this has been clarified within the text.

Section: Materials & Methods

Line numbers: 134

  1. It might be interesting to add the questionnaires used in English as supplementary material.

To avoid copyright issues with some scales that are not freely available and to maintain conciseness, the authors have decided not to include all the scales within the supplementary material. Readers can access further information about the scales via the citations included in the methods section.

Reviewer 2 Report

Why were traditional questionnaires not used – weaker because you are not using validated and reliability tested measures – far too many scales used why?

There is far too much information in this report – it is too dense to discern what is important.

The analyses was well thought out but the reader is inundated with so many tables and statistics that the paper doesn’t appear to making any particular point.  

How correlated are these measures? Can you take some out?  The paper loses impact due to so many results that are essentially saying the same thing,

Author Response

Reviewer 2

  1. Why were traditional questionnaires not used – weaker because you are not using validated, and reliability tested measures – far too many scales used why?

The only measures that were not validated were the COVID-19 questions. These were not validated, as, at the time of testing, COVID-19 had only recently been announced as a pandemic. As such, there were no validated COVID-19 measures at that point in time due to the novelty of the pandemic. We have included a statement in the Materials & Methods to clarify this. Our psychological measures; however, were all established measures in the literature and we provided respective reliability coefficients for our samples. 

Section: Materials & Methods

Line numbers: 122 – 123

  1. There is far too much information in this report – it is too dense to discern what is important.

We understand that there was a lot of measures included in this paper. We believe it provided a comprehensive overview of the current mental health and wellbeing of the students that participated in our studies. We did not want to selectively report only some measures, as we had no rationale to pick and choose what would be important but rather selected to be transparent in the measures we had data for that were comparative between countries. With this reviewer’s comment in mind, we have moved some tables into the supplementary materials to de-bulk the paper and improve its readability.

Section: Results

Line numbers: 671 - 732

  1. The analyses was well thought out but the reader is inundated with so many tables and statistics that the paper doesn’t appear to making any particular point.  

Similar to the previous response, we have moved some tables to the supplementary materials to improve the readability of the manuscript. We have selected to report the data as transparent and comprehensive as possible to reduce any selection bias.

Section: Results

Line numbers: 671 - 732

  1. How correlated are these measures? Can you take some out?  The paper loses impact due to so many results that are essentially saying the same thing,

Although some of the scales measuring similar constructs are correlated, the authors believe it is important for transparency in research, to report all the measures that were collected in both studies to highlight the overall mental well-being of teens by asking questions that could capture their reality. We would not have any strong argument/rationale to selectively remove some measures more than others. However, we agree with the reviewer that there is a lot of information being presented and have moved some of the larger tables to the supplementary materials to improve the readability of the paper.

Section: Results

Line numbers: 671 - 732

Reviewer 3 Report

This manuscript is very nicely done. I just a few suggestions/requests:

  1. Please note that #5 affiliation is missing.
  2. Towards the end of the abstract, ..."the pandemic likely has.." the has should be have. I would check proper use of singular/plural throughout the manuscript as I saw the incorrect use another time as well.
  3. Figure 1 - I don't think pie charts are the best to show this data. I think it should actually just be in the text.
  4. Figure 2 - The points and CIs should not be connected with a line. Unless I am missing something here, these are separate measures and not a continuous trend, therefore connecting them doesn't make sense. This is my most important suggested change.

Nice work on this, I think it will contribute to the literature.

Author Response

Reviewer 3

This manuscript is very nicely done. I just a few suggestions/requests:

  1. Please note that #5 affiliation is missing.

The authors thank the reviewer for highlighting this error and have resolved this in-text.

Section: Title page

Line numbers: 11

  1. Towards the end of the abstract, ..."the pandemic likely has.." the has should be have. I would check proper use of singular/plural throughout the manuscript as I saw the incorrect use another time as well.

The authors thank the reviewer for picking up these grammatical errors, which have now been amended in-text.

Section: Abstract, Discussion

Line numbers: 28, 368

  1. Figure 1 - I don't think pie charts are the best to show this data. I think it should actually just be in the text.

The authors would like to retain the pie charts, as it provides a more concise and consistent way of communicating and visualising the percentage of students experiencing symptoms of depression from the two different depression measures utilised in the studies.

Section: Results

Line numbers: 332 – 338

  1. Figure 2 - The points and CIs should not be connected with a line. Unless I am missing something here, these are separate measures and not a continuous trend, therefore connecting them doesn't make sense. This is my most important suggested change.

Thank you for highlighting this point, we have removed the lines connecting the points.

Section: Results

Line numbers: 351 - 354

Nice work on this, I think it will contribute to the literature.